# HARMONYLM: ADVANCING UNIFIED LARGE-SCALE LANGUAGE MODELING FOR AUDIO AND MUSIC GENERATION

## ABSTRACT

The fields of sound generation and music generation have seen notable advancements with the development of specialized models tailored to each domain. However, these domains share commonalities, and the use of specialized models can lead to increased hardware resource requirements. On the other hand, recent breakthroughs in large language models, particularly in natural language processing, have showcased their ability to capture complex patterns and generate coherent and contextually relevant outputs in various tasks. Leveraging the success of these language models, we present HarmonyLM, a unified framework designed to synthesize sound and music from discrete representations. HarmonyLM adopts a unified perspective in modeling sound and music, discrete tokens are modeled from text descriptions using a decoder-only model, which are converted back to harmonious and consistent audio outputs. HarmonyLM offers significant advantages as a unified sound and music generation framework. (1) Model Scalability: the model we use in acoustic modeling a decoder-only transformer, which is free to scale up model size. (2) Data Scalability: the acoustic modeling and reconstructing audio models do not require any annotations, which accommodate different scales of data. Experimental results demonstrate the effectiveness of HarmonyLM, as it achieves superior audio quality compared to competitive baseline models.[1]

## 1 INTRODUCTION

Text-to-sound/music (Huang et al., 2023c; Copet et al., 2023; Huang et al., 2023a) aims to generate high-quality audios given text descriptions, which makes significant progress with the development of deep generative models like diffusion models (Ho & Salimans, 2021; Yang et al.). They are of importance for live dubbing, game sound effects, film and television soundtracks, and virtual reality(VR) and augmented reality(AR) applications. Despite significant success being made in developing specialized models for sound and music generation, separate models limit their ability to cope with more complex auditory scenarios. For example, battle scenes in games or movies require sound effects such as fighting sounds and explosions to simulate real scenes, and appropriate music adds to the audience's emotional experience and helps establish the atmosphere of the battle scene.

Large language models(LLM) (Touvron et al., 2023a;b; Zeng et al., 2022) have achieved great success in natural language processing(NLP), which proves that large language models can learn more complex paradigms. Recent advancements in self-supervised audio representation learning (Zeghidour et al., 2021; Défossez et al., 2022), sequential modeling (Yu et al., 2023; Brown et al., 2020), and audio synthesis (Borsos et al., 2022; Agostinelli et al., 2023) provide the conditions to develop a large-scale language model framework for music and sound generation. To make audio modeling more tractable, recent studies like SoundStream (Zeghidour et al., 2021) and Encodec (Défossez et al., 2022) proposed representing audio signals as multiple streams of discrete tokens representing the same signal. This allows both high-quality audio generation and effective audio modeling. Current large-scale audio generation systems (Kharitonov et al., 2023; Shen et al., 2023) leverage the codec models to generate discrete tokens and then predict them using language models, which have addressed intelligibility challenges in generated samples through large-scale training.

---

[1]Audio samples are available at `https://HarmonyLM.github.io`

While previous sound or music generation models have proven their effectiveness in their respective fields, most of them have been developed independently despite generating "audio" as a common objective. Thus, the methodologies developed for each application remain scattered in research fields, which is inefficient since we still need to optimize separated models for sound or music generation tasks.

In this work, we introduce HarmonyLM, a unified sound and music framework for synthesizing high-quality audio from discrete representations. HarmonyLM takes a unified perspective to model sound and music, which model discrete tokens from text description with a decoder-only language model, and then map them back to high-fidelity waveforms from discrete tokens. HarmonyLM demonstrates notable advantages as a unified audio synthesis framework: 1) Template universality: we define a novel prompt template to combine any texts and discrete units. 2) Model scalability: the acoustic backbone adopts a variant of transformer decoders, and thus the model capacity could be scaled up. Experimental results demonstrate that HarmonyLM achieves new state-of-the-art results. Both subjective and objective evaluation metrics show that HarmonyLM exhibits superior audio quality and style similarity compared with baseline models.

Our contributions can be summarized as follows:

- We propose a unified sound and music generation model called HarmonyLM, which effectively generates consistent and harmonious audio with a language model based on text prompts.
- We investigate the model and data scalability if large language models for both sound and music generation.
- Experimental results on two tasks demonstrate that HarmonyLM achieves state-of-the-art results in terms of subjective and objective metrics. We demonstrate our scalability by conducting experiments with different capacity language models and text encoders.

## 2 RELATED WORKS

### 2.1 TEXT-GUIDED SOUND/MUSIC GENERATION

Text-to-sound and text-to-music generation exhibit commonalities. However, previous works (Huang et al., 2023c; Liu et al.; Dhariwal et al., 2020) often design task-specific inductive biases that limit their generalizability. One popular approach is the use of diffusion models, which naturally operate on continuous representations. Diffsound (Yang et al., 2022), for instance, leverages a pre-trained VQ-VAE (van den Oord et al., 2018) on mel-spectrograms to convert sound into discrete codes. These codes are then utilized by a diffusion model to generate audio outputs. Schneider et al. (2023); Huang et al. (2023b); Maina (2023) proposes the use of latent diffusion models for text-to-music while (Huang et al., 2023a; Liu et al.; Ghosal et al., 2023) for text-to-sound. Alternatively, music or sound samples can be represented as discrete codes using a hierarchical VQ-VAE, allowing the construction of language models on top of it. For example, AudioGen (Kreuk et al.) and MusicGen (Copet et al., 2023) encode raw waveform data into discrete codes and employ auto-regressive models to predict audio tokens based on text features. To advance towards a unified perspective of sound and music generation, this work presents a novel unified framework that capitalizes on a language model for both music and sound. By leveraging the power of language models, this framework seeks to facilitate the generation of coherent and expressive audio outputs in various applications.

### 2.2 AUDIO REPRESENTATION

In recent years, there has been a significant increase in research dedicated to the compression of audio signals into continuous discrete representations. This approach aims to achieve efficient speech processing and high-fidelity audio coding. Notable advancements in this field include Wav2Vec (Baevski et al., 2020) and Hubert (Hsu et al., 2021), which have proposed quantization techniques using k-means to compress speech representations effectively. Drawing inspiration from vector quantization (VQ), SoundStream (Zeghidour et al., 2021) and Encodec (Défossez et al., 2022) have explored the utilization of hierarchical architectures to represent acoustic information. These models offer promising solutions for capturing and reconstructing audio signals with improved quality and fidelity. In a recent study by (Yang et al., 2023), they introduced a novel technique known as group-residual vector quantization (GRVQ), which demonstrates enhanced performance in audio

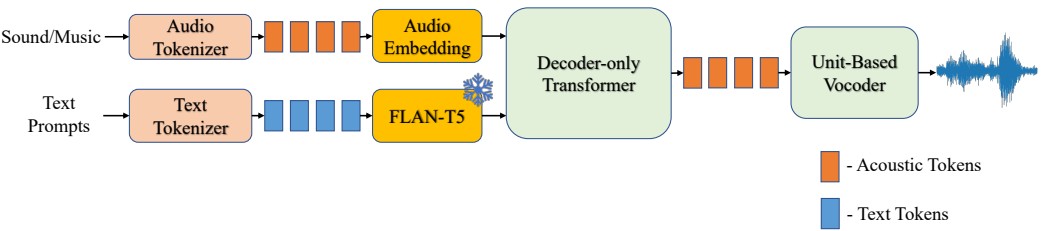

Figure 1: A high-level overview of HarmonyLM. We freeze the FLAN-T5 as our text encoder and use SoundStream as the audio tokenizer. The generated acoustic tokens by the autoregressive transformer are converted back to the raw sound/music with the unit-based vocoder(see Section 3.5).

coding. Based on this breakthrough, our work builds upon the techniques of SoundStream to extract discrete representations as acoustic tokens from sound and music. This enables us to achieve effective audio synthesis and processing, further enhancing the capabilities of our proposed framework.

## 2.3 LANGUAGE MODELS

In the realm of audio synthesis, there has been a growing interest in modeling audio signals within a compact and discrete space. This approach allows for efficient and effective representation of audio using autoregressive transformers. Pioneering works such as AudioLM (Borsos et al., 2022) and MusicLM (Agostinelli et al., 2023) view audio synthesis as a language modeling task and employ a hierarchical structure of coarse-to-fine units. By leveraging this hierarchical approach, these models can generate high-quality audio outputs with fine-grained control. SpeechDLM (Nguyen et al., 2023) takes a similar approach but focuses specifically on spoken language modeling for dialogue. By utilizing the HuBERT representation, SpeechDLM introduces an end-to-end framework for generating realistic and context-aware speech. Furthermore, recent advancements in the field, such as MusicGen (Copet et al., 2023), propose operating over multiple streams of compressed discrete music representations. This allows for the synthesis of complex and expressive musical compositions. In this study, we propose a unified framework for sound and music generation. This framework adopts an autoregressive sequence-to-sequence (seq2seq) approach and utilizes discrete representations.

## 3 HARMONYLM

### 3.1 OVERVIEW

HarmonyLM stands as a unified framework that seamlessly combines language modeling and discrete representations to facilitate text-to-sound and text-to-music synthesis. As depicted in Figure 1. HarmonyLM employs autoregressive generation to transform audio samples into discrete tokens, which enables the model to capture the intricate acoustic characteristics and nuances embedded within the provided text prompts. By effectively mapping text prompts to corresponding acoustic tokens, HarmonyLM establishes a strong foundation for subsequent audio reconstruction. In the following, the model maps the acoustic token back to audio using a unit-based vocoder.

Once the backbone network of HarmonyLM has been trained, the model can be applied to two distinct domains: text-to-sound and text-to-music generation. Both tasks can efficiently utilize a shared language model with discrete representations. Flan-T5, a powerful text encoder, is employed in both domains to effectively encode the textual prompts. The proposed unified prompt template is then leveraged to seamlessly combine the text prompts with the acoustic tokens, facilitating a cohesive integration of textual and acoustic information. Following the combination of text prompts and acoustic tokens, acoustic tokens are generated and reconstructed into audio.

### 3.2 DISCRETE AUDIO REPRESENTATION

Audio codec models such as SoundStream (Zeghidour et al., 2021) and Encodec (Défossez et al., 2022) have recently shown that encoder-decoder architecture excels at learning acoustic information in a self-supervised manner, where the representation can be used in a variety of generative tasks.

The acoustic codec model typically consists of an audio encoder, a residual vector-quantizer (RVQ), and an audio decoder: 1) The audio encoder $E$ consists of several convolutional blocks with a total downsampling rate of 320 and generates continuous representations at every 20-ms frame in 16kHz. 2) The residual vector-quantizer $Q$ produces discrete representations $a_q$ with a codebook size of $K_2$, using a vector quantization layer (Vasuki & Vanathi, 2006). 3) The audio decoder $G$ reconstructs the signal $\hat{y}$, from the compressed latent representation $a_q$.

### 3.3 TEXTUAL REPRESENTATION

Text-guided synthesis models require powerful semantic text encoders to effectively capture the meaning of arbitrary natural language inputs. These encoders can be categorized into two major groups: 1) Contrastive pretraining. Similar to CLIP (Radford et al., 2021) pre-trained on image-text data, recent progress on contrastive language-audio pretraining (CLAP) (Elizalde et al.) brings audio and text descriptions into a joint space and demonstrates the outperformed zero-shot generalization to multiple downstream domains. 2) Large-scale language modeling (LLM). Saharia et al. and Kreuk et al. (2023) utilize language models (e.g., BERT (Devlin et al., 2018), T5 (Raffel et al.), FLAN-T5 (Chung et al., 2022)) for text-guided generation. Language models are trained on text-only corpus significantly larger than paired multi-modal data, thus being exposed to a rich distribution of text.

The FLAN-T5 models have undergone pre-training on a large-scale chain-of-thought (CoT) and instruction-based dataset. This extensive pre-training enables the models to learn new tasks effectively by leveraging in-context information and mimicking gradient descent through attention weights. This property is not present in older large models or contrastive models like T5 and CLAP. Considering the advantages mentioned above, we opt to utilize the pre-trained FLAN-T5-LARGE model as our text encoder, which is frozen following the common practice (Ghosal et al., 2023; Ramesh et al., 2022).

### 3.4 ACOUSTIC MODELING

#### 3.4.1 UNIFIED PROMPT TEMPLATE

To facilitate a wide range of text-guided tasks, our model has been designed to handle various combinations of text prompts and discrete units, denoted as <Text, Acoustic>. Drawing inspiration from the patch embedding technique used in vision transformers (Dosovitskiy et al., 2021), we chunk the input sequences into P patches, where P corresponds to the number of quantization levels.

To represent the text prompts, we introduce a special token ([Continuous Token]) and replicate it P times. These replicated tokens are then concatenated with the corresponding acoustic tokens. Subsequently, the continuous tokens are replaced with text features after undergoing a patch embedding process. To differentiate between the two types of sequences (text and acoustic), we utilize two special tokens ([TYPE_START] and [TYPE_END]), where TYPE refers to the type of sequence being processed. The unified prompt template can be defined as follows:

$$x_p = [\text{T5\_Start}] \, C \, [\text{T5\_End}] \, [\text{Acoustic\_Start}] \, A \, [\text{Acoustic\_End}] \tag{1}$$

where C denotes continuous tokens and A is acoustic tokens, and all special tokens repeat P times.

#### 3.4.2 DECODER-ONLY TRANSFORMER

After constructing a unified input template, a decoder-only transformer is adopted to map text prompts into acoustic tokens. To achieve a unified large-scale language model for audio and music generation and leverage the capabilities of language models for reconstructing acoustic tokens, we employ the following strategies:

- Model Scalability: The transformer model, renowned for its parallel computation and self-attention mechanisms, enables the handling of large volumes of input and output data. This design choice allows our model to efficiently scale up to larger capacities, accommodating the demands of more complex sound and music generation tasks.

- Data Scalability: To enhance data scalability, we utilized discrete representations for modeling sound and music. In the acoustic modeling stage, we employ a decoder-only model

to generate discrete units from textual descriptions. This discrete representation not only reduces storage and transmission costs but also improves training and inference efficiency. By adopting discrete representations, our framework eliminates the need for annotated data and achieves better handling of large-scale sound and music data, facilitating sophisticated modeling and generation of sound and music.

These approaches enable us to realize a unified framework for sound and music generation, leveraging the capabilities of language models for acoustic unit reconstruction. The model scalability empowers us to handle larger and more complex tasks, while the data scalability optimizes the handling of large-scale sound and music datasets, enhancing efficiency and accuracy in modeling and generation.

### 3.5 RECONSTRUCTING AUDIO

Upon completion of the training process, we can utilize language models to generate acoustic tokens based on the given text prompts. Subsequently, a unit-based vocoder is employed to synthesize the corresponding audio waveforms. It is worth noting that the acoustic codec used, such as SoundStream, leverages multiple quantization levels, typically 12, to enhance the quality of audio reconstruction. Thus, reducing the number of codebooks during the inference stage might result in a noticeable drop in perceptual quality.

To ensure that the quality of the generated audio waveforms remains uncompromised, we adopt a unit-based neural vocoder that is trained from scratch for waveform generation from acoustic units. This vocoder achieves high-quality audio reconstruction using only three quantization levels. Taking inspiration from the BigVGAN model (Lee et al., 2022), our synthesizer consists of a generator and a multi-resolution discriminator (MRD). The generator incorporates a set of look-up tables (LUT) that embed the discrete representations, along with a series of blocks. Each block comprises transposed convolutions and a residual block with dilated layers. The transposed convolutions upsample the encoded representation to match the input sample rate, while the dilated layers increase the receptive field.

### 3.6 TRAINING AND INFERENCE PROCEDURES

#### 3.6.1 TRAINING

During the training of language models, we compute the cross-entropy (CE) loss between the generated and target units. In the phase of audio reconstruction, we train the enhanced vocoder using a combination of different loss functions. These include the least-square adversarial loss, the feature matching loss, and the spectral regression loss on mel-spectrograms. These loss functions are carefully weighted and summed together, following the formulations and hyperparameters established by previous works such as Kong et al. (2020a); Lee et al. (2022)

The training data can be scaled up to a large-scale dataset, facilitating the modeling and generation of sound and music in a sophisticated manner. Moreover, HarmonyLM leverages the inherent scalability of transformer models. This scalability enables us to adapt the framework to different model sizes, accommodating the demands of more complex sound and music generation tasks.

#### 3.6.2 INFERENCE

HarmonyLM exhibits efficient advantages as a unified audio framework with discrete tokens and language modeling. Text-to-sound and text-to-music can be tackled by generating acoustic representations with text prompts: sound or music sample is tokenized into acoustic tokens, and unified prompt template are applied to combine text prompts and acoustic tokens.

## 4 EXPERIMENTS

### 4.1 EXPERIMENTAL SETUP

#### 4.1.1 DATA

For training text-to-sound models, we use a combination of several datasets: AudioSet, BBC sound effects, Audiostock, AudioCaps-train, ESC-50, FSD50K, Free To Use Sounds, Sonniss Game Effects, WeSoundEffects, MACS, Epidemic Sound, UrbanSound8K, WavText5Ks, LibriSpeech, and Medley-solos-DB. For audios without natural language annotation, we apply the pseudo prompt enhancement to construct captions aligned well with the audio. Overall we have ∼3k hours with 1M audio-text pairs for training data. For evaluating text-to-sound models (Yang et al., 2022; Kreuk et al., 2023), the AudioCaption validation set is adopted as the standard benchmark, which contains 900 samples with five human-annotated captions in each audio clip. For a more challenging zero-shot scenario, we also provide results in the Clotho (Drossos et al.) validation set which contains multiple audio events.

For training text-to-music models, we use the LP-MusicCaps-MSD, which includes 500,000 pieces of music with 2.2M Caption. We evaluate the text-to-music models on the LP-MusicCaps-MC evaluation set. We convert the sampling rate of all audios to 16kHz and set the maximum length of the text to 77.

#### 4.1.2 MODEL CONFIGURATIONS

For acoustic representation, we train the SoundStream model with 12 quantization levels, each with a codebook of size 1024 and the same downsampling rate of 320. We take 3 quantization levels as the acoustic tokens, representing each frame as a flat sequence of tokens from the first, second, and third quantization layers.

We use FLAN-T5-Large as our text encoder. Autoregressive acoustic modeling global models are 30-layer transformers with an attention dimension of 1920 and an FFN dimension of 7680. As for the unit-based vocoder, we use the modified V1 version of BigVGAN. A comprehensive table of hyperparameters is available in Appendix B.

#### 4.1.3 TRAINING AND EVALUATION

During training, we train acoustic modeling transformers respectively for 50K steps using 8/80 NVIDIA A100 GPUs with a batch size of 10000 tokens for each GPU on the publicly-available *fairseq* framework (Ott et al., 2019). Adam optimizer is used with $\beta_1 = 0.9, \beta_2 = 0.98, \epsilon = 10^{-9}$. For the acoustic modeling, we crop the waveform to a random length of up to 10 seconds. Reconstructing audio model is optimized with a segment size of 8192 and a learning rate of $1 \times 10^{-4}$ until 500K steps using 4 NVIDIA A100 GPUs. During inference, we use batch size 1 of autoregressive decoding in acoustic modeling.

#### 4.1.4 EVALUATION METRICS

To evaluate the performance of HarmonyLM on text-to-sound and text-to-music tasks, we follow the common practice of Huang et al. (2023a) and Copet et al. (2023). Specifically, to evaluate the text-to-sound models, we include both objective metrics Frechet distance (FD), Kullback–Leibler (KL) divergence, Frechet audio distance (FAD), and CLAP score, and subjective metrics including MOS-Q and MOS-F to assess the audio quality and the text-audio alignment faithfulness. The FAD is a reference-free perceptual metric that measures the distance between the generated and ground truth audio. FD measures the similarity between the generated and ground truth audio samples while CLAP score is a reference-free metric to measure audio-text alignment. As for subjective evaluation, we leverage the Amazon Mechanical Turk, a crowd-sourced platform, to perform the subjective evaluation on metrics including MOS-Q and MOS-F. We use similar evaluation metrics except for FD score, following Copet et al. (2023). More information regarding the evaluation process can be found in Appendix C.2

| Model | Params | FD↓ | KL↓ | FAD↓ | CLAP↑ | MOS-Q↑ | MOS-F↑ |
|---|---|---|---|---|---|---|---|
| GroundTruth | - | - | - | - | 0.671 | 86.47 | 84.31 |
| AudioGen-S | 285M | - | 2.09 | 3.13 | - | - | - |
| AudioGen-L | 1B | - | 1.69 | 1.82 | - | - | - |
| Make-An-Audio | 453M | 18.32 | 1.61 | 2.66 | 0.593 | 69.54 | 65.45 |
| AudioLDM-S | 454M | 29.48 | 1.97 | 2.43 | - | - | - |
| AudioLDM-L | 1.01B | 23.31 | 1.59 | 1.96 | 0.605 | 70.91 | 67.41 |
| TANGO | 1.21B | 26.13 | 1.37 | 1.87 | **0.650** | 74.10 | 72.76 |
| Make-An-Audio 2 | 937M | 15.25 | 1.32 | 1.80 | 0.645 | 78.31 | 75.63 |
| HarmonyLM | 1.3B | **12.5** | 1.82 | **1.49** | 0.43 | **80.01** | **76.22** |

Table 1: The comparison between HarmonyLM and baseline models on the AudioCaps dataset. All the diffusion-based models run with 100 DDIM steps for a fair comparison. We used the model released by the authors on Huggingface to test scores.

## 4.2 TEXT-TO-SOUND

We compare the generated audio samples with several popular audio generation systems, including 1) GT, the ground-truth audio; 2) AudioGen (Kreuk et al., 2023); 3) Make-An-Audio (Huang et al., 2023c); 4) AudioLDM (Liu et al.); 5) TANGO (Ghosal et al., 2023); 6) Make-An-Audio 2 (Huang et al., 2023a); For easy comparison, the results are compiled and presented in Table 1, and we have the following observations: 1) For the quality of generated sounds, HarmonyLM has achieved higher FAD scores and FD scores than all baselines, both latent diffusion models(LDM) and language models, indicating that HarmonyLM can generate accessible sound of good quality as most previous LDM families. 2) Regarding subjective evaluation results, HarmonyLM has achieved the highest MOS-Q of 80.01 and MOS-F of 76.22 compared with the baseline models, demonstrating that HarmonyLM can better simulate real-world sounds. 3) For text-to-audio similarity, although HarmonyLM has a relatively lower CLAP score compared to the LDM benchmark, we argue this is mainly due to the difference caused by HarmonyLM not using classifier-free guidance and other data augmentation methods for text.

| Model | Clotho-eval | | | AudioCaps-test | | |
|---|---|---|---|---|---|---|
| | FD↓ | KL↓ | FAD↓ | FD↓ | KL↓ | FAD↓ |
| TANGO | 32.1 | 2.59 | 3.61 | 31.76 | 2.04 | 10.53 |
| AudioLDM-L | 28.15 | 2.6 | 4.93 | 31.97 | 2.39 | 6.79 |
| Make-An-Audio 2 | 22.79 | **2.52** | 2.76 | 13.78 | **1.61** | 2.33 |
| HarmonyLM | **20.96** | 2.80 | **2.64** | **12.5** | 1.82 | **1.49** |

Table 2: Comparison of HarmonyLM, Make-An-Audio 2, AudioLDM-L, and Tango on Clotho-eval and AudioCaps-test datasets.

**Zero-shot evaluation.** As illustrated in Table 2, when migrating to a more challenging scenario to Clotho in a zero-shot fashion, HarmonyLM still exhibits better FD and FAD scores and comparable KL scores, demonstrating HarmonyLM's effectiveness in constructing diverse object compositions for better generalization.

## 4.3 TEXT-TO-MUSIC

In this part, we compare the generated audio samples with other systems, including 1) GT, the ground-truth audio; 2) MusicGen (Copet et al., 2023); 3) MusicLM (Agostinelli et al., 2023); 4) Mousai (Schneider et al., 2023); 5) Riffusion (Forsgren & Martiros, 2022). The results are presented in Table 3, and we have the following observations: 1) Compared with diffusion-based models like Mousai and Riffusion, HarmonyLM performs better in both objective and subjective metrics, especially in the fact that the FAD score dropped from 7.5 to 2.95. This demonstrates the potential of language models in the field of music generation to synthesize harmonious and consistent pieces of music.

| Model | KL↓ | FAD↓ | CLAP↑ | MOS-Q↑ | MOS-F↑ |
|---|---|---|---|---|---|
| GroundTruth | - | - | 0.40 | 88.42 | 90.34 |
| Riffusion | 2.06 | 14.8 | 0.19 | 79.31 | 74.20 |
| Mousai | 1.59 | 7.5 | 0.23 | 76.11 | 77.35 |
| MusicLM | - | 4.0 | - | 80.51 | 82.35 |
| MusicGen | **1.23** | 3.4 | 0.32 | 80.74 | 83.70 |
| HarmonyLM | 1.50 | **2.95** | **0.34** | **82.32** | **85.28** |

Table 3: The comparison between HarmonyLM and baseline models on the MusicCaps Evaluation set. We borrow the results from the MusicGen (Copet et al., 2023).

2) Compared with language models like MusicLM and MusicGen, HarmonyLM surpasses MusicLM in FAD score, improving from 4.0 to 2.95, and outperforms MusicGen in FAD and CLAP. The MOS-Q and MOS-F metrics have achieved the highest scores and exceed MusicLM by 1.81 and 2.93 respectively. This shows that HarmonyLM as an efficient unified model is able to generate high-quality music as the separate models do, which saves computing resources and time.

## 4.4 ANALYSIS AND ABLATION STUDIES

To verify the effectiveness of several designs in HarmonyLM, including the scalability of language models and text representations, we conduct ablation studies and discuss the key findings as follows.

| Model | Params | FD↓ | KL↓ | FAD↓ | CLAP↑ |
|---|---|---|---|---|---|
| **Model Scalablity** | | | | | |
| HarmonyLM-S | 400M | 14.1 | 2.36 | 1.79 | 0.39 |
| HarmonyLM-M | 850M | 13.2 | 2.01 | 1.6 | 0.41 |
| HarmonyLM-L | 1.3B | 12.5 | 1.82 | 1.49 | 0.43 |
| HarmonyLM-XL | 2.2B | 12.0 | 1.73 | 1.43 | 0.44 |
| **Data Scalablity** | | | | | |
| HarmonyLM-AC | 1.3B | 14.0 | 1.95 | 1.87 | 0.36 |
| HarmonyLM-ALL | 1.3B | 12.0 | 1.73 | 1.43 | 0.44 |
| **Text Representions** | | | | | |
| T5-Large | 400M | 18.35 | 3.15 | 2.73 | 0.30 |
| FLAN-T5-Base | 400M | 14.65 | 2.42 | 1.90 | 0.37 |
| FLAN-T5-Large | 400M | 14.1 | 2.36 | 1.79 | 0.39 |
| FLAN-T5-XL | 400M | 13.85 | 2.13 | 1.71 | 0.41 |

Table 4: We conduct ablation studies on the text-to-sound task with different model sizes and text encoder dimensions. HarmonyLM-AC denotes training HarmonyLM on AudioCaps while Harmony-ALL on all datasets.

### 4.4.1 IMPACT OF ARCHITECTURE SCALE

We investigate the impact of the model size on the text-to-sound task performance. We train four HarmonyLM models of different sizes(400M, 800M, 1.3B, 2.2B) using discrete representations. As illustrated in Table 4, we find that all metric scores improve substantially with model size, with a 2.1 reduction in FD and a 0.36 drop in FAD moving from 400M to 2.2B model. This proves that larger language models have better understanding and audio reconstruction capabilities while requiring more computing resources and training time.

### 4.4.2 SCALING THE TRAINING DATA.

We investigate the impact of increasing the amount of training data on text-to-sound task performance. We run this analysis on HarmonyLM 1B checkpoint and train this model with an increasing amount

of data: (1) The AudioCaption set only. (2) All the datasets described in Section A. We can draw a conclusion from the observation of Table 4. We find that training with increasing amounts of data yields a substantial improvement which demonstrates the advantage of HarmonyLM in terms of data scalablity.

### 4.4.3 TEXTUAL REPRESENTATION

We explore different text representations including T5-Large, FLAN-T5-Base, FLAN-T5-Large, and FLAN-T5-XL in the text-to-sound task. We freeze the weights of text encoders. For easy comparison, we present the results in Table 4 and have the following observations: 1) A larger text encoder produces higher scores across all metrics, which shows that higher feature dimensions could provide richer information representation and more powerful semantic modeling capabilities. By increasing feature dimensions, text encoders can better capture subtle features and semantic information in text input, thereby improving model performance. 2) FLAN-T5-Large outperforms T5-Large across all metrics especially FD score and FAD score, which demonstrates FLAN-T5 are able to learn a new task better with pre-training of FLAN-T5 models on a large-scale chain-of-thought- (CoT) and instruction-based dataset.

## 5 CONCLUSION

In this work, we proposed HarmonyLM, a unified model for synthesizing sound and music from discrete representations. HarmonyLM adopted a unified perspective in modeling sound and music: discrete tokens are modeled from text descriptions using a decoder-only model, which are mapped back to audio using codec models for harmonious and consistent audio generation. Experimental results demonstrated that HarmonyLM offered significant advantages as a unified sound and music generation framework: (1) Model Scalability: the model we use in acoustic modeling are decoder-only transformer, which is free to scale up model size. (2) Data Scalability: the acoustic modeling and reconstructing audio models do not require any annotations, which accommodate different scales of data. For future work, we will verify the effectiveness in more general scenarios such as audio generalization. The discussions on limitations and potential negative impacts are included in the Appendix.

## 6 LIMITATION

HarmonyLM utilizes auto-regressive models for the unified generation of sound and music, which inherently involves an iterative refinement process to achieve better results. The model's ability to produce high-quality outputs is directly influenced by factors such as sequence length and available computational resources. Longer sequence lengths typically require more computational power, and there may be a degradation in performance when training data is limited. One of our future directions is to develop lightweight and parallel models for accelerating sampling.

Furthermore, it is important to acknowledge that HarmonyLM currently lacks certain strategies such as classifier-free guidance and large language model data augmentation, which have been employed in previous latent diffusion models. These strategies play a crucial role in aligning generated audio and text, ensuring coherence and fidelity. In future work, we plan to incorporate these techniques into HarmonyLM to further improve its performance.

## 7 POTENTIAL NEGATIVE SOCIETAL IMPACTS

HarmonyLM lowers the requirements for harmony and consistent sound and music generation, which may cause unemployment for people with related occupations such as voice actor and composer. In addition, there is the potential for harm from non-consensual sound or music generation of fake media might be over-used than they expect.

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

## A  DATASET

| Dataset | Hours | Type | Source |
|---|---|---|---|
| Clotho | 152 | Caption | Drossos et al. |
| AudioCaps | 109 | Caption | Kim et al. |
| MACS | 100 | Caption | Martín-Morató & Mesaros |
| WavText5Ks | 25 | Caption | Deshmukh et al. (2022) |
| BBC sound effects | 481 | Caption | `https://sound-effects.bbcrewind.co.uk/` |
| Audiostock | 43 | Caption | `https://audiostock.net/se` |
| Filter AudioSet | 2084 | Label | Gemmeke et al. |
| ESC-50 | 3 | Label | Piczak |
| FSD50K | 108 | Label | `https://annotator.freesound.org/fsd/` |
| Sonniss Game Effects | 20 | Label | `https://sonniss.com/gameaudiogdc/` |
| WeSoundEffects | 11 | Label | `https://wesoundeffects.com/` |
| Epidemic Sound | 220 | Label | `https://www.epidemicsound.com/` |
| UrbanSound8K | 8 | Label | Salamon et al. |
| LibriTTS | 300 | Language-free | Zen et al. (2019) |
| LP-MusicCaps-MSD | 4283 | Caption | Doh et al. (2023) |
| LP-MusicCaps-MC | 15 | Caption | Doh et al. (2023) |

Table 5: Statistics for the combination of several datasets.

As shown in Table A, we collect a large-scale sound-text dataset consisting of 1M audio samples with a total duration of ∼3k hours. It contains audio of human activities, natural sounds, and audio effects, consisting of several data sources from publicly available websites. For audio with text descriptions, we download the parallel audio-text data. For audios without natural language annotation (or with labels), we discard the corresponding class label (if any) and apply the pseudo prompt enhancement to construct natural language descriptions aligned well with the audio. As for the music dataset, we collect a large-scale music-text dataset consisting of 0.5M music samples with 2.2M captions.

## B  MODEL CONFIGURATIONS

We list the model hyper-parameters of HarmonyLM in Table 6.

| Hyperparameter | | HarmonyLM |
|---|---|---|
| | Transformer Layer | 24 |
| | Transformer Embed Dim | 1920 |
| | Transformer Attention Headers | 16 |
| Decoder-Only Transformer | Transformer FFN Embed Dim | 7680 |
| | Decoder Dictionary Length | 3081 |
| | Number of Parameters | 1.3B |
| | Upsample Rates | [5, 4, 2, 2, 2, 2] |
| Vocoder | Hop Size | 320 |
| | Upsample Kernel Sizes | [9, 8, 4, 4, 4, 4] |
| | Number of Parameters | 121.6M |
| Total Number of Parameters | | 1.4B |

Table 6: Hyperparameters of HarmonyLM.

We list our architecture scale in Table **??**

## C  UNIT-BASED VOCODER

The generator of the unit-based vocoder is built from a set of look-up tables (LUT) that embed the discrete representation, and a series of blocks composed of transposed convolution and a residual block with dilated layers.

| Params | Layer | Head | Hidden Dim |
|--------|-------|------|------------|
| 400M   | 26    | 24   | 1152       |
| 850M   | 30    | 24   | 1536       |
| 1.3B   | 30    | 32   | 1920       |
| 2.2B   | 30    | 40   | 2304       |

Table 7: Different model scale of HarmonyLM.

# D EVALUATION

## D.1 SUBJECTIVE EVALUATION

To assess the generation quality, we conduct MOS (Mean Opinion Score) tests regarding audio quality and text-audio faithfulness, respectively scoring MOS-Q and MOS-F.

For audio quality, the raters were explicitly instructed to "focus on examining the audio quality and naturalness." The testers were presented with audio samples and asked to rate their subjective score (MOS-P) on a 20-100 Likert scale.

For text-audio faithfulness, human raters were shown the audio and its caption and asked to respond to the question, "Does the natural language description align with the audio faithfully?" They had to choose one of the options - "completely," "mostly," or "somewhat" on a 20-100 Likert scale.

Our crowd-sourced subjective evaluation tests were conducted via Amazon Mechanical Turk where participants were paid $8 hourly. A small subset of the generated audio samples used in the test can be found at `https://HarmonyLM.github.io/`.

## D.2 OBJECTIVE EVALUATION

Fréchet Audio Distance (FAD) (Kilgour et al., 2018) is adapted from the Fréchet Inception Distance (FID) to the audio domain, it is a reference-free perceptual metric that measures the distance between the generated and ground truth audio distributions. FAD is used to evaluate the quality of generated audio.

KL divergence is measured at a paired sample level between the generated audio and the ground truth audio, it is computed using the label distribution and is averaged as the final result.

Fréchet Distance (FD) evaluates the similarity between the generated and ground truth audio distributions. FD, KL and IS are built upon an audio classifier, PANNs (Kong et al., 2020b), which takes the mel-spectrogram as model input. Differently, FAD uses VGGish (Hershey et al., 2017) as an audio classifier that takes raw audio waveform as model input.

CLAP score: adapted from the CLIP score (Hessel et al., 2021; Radford et al., 2021) to the audio domain and is a reference-free evaluation metric to measure audio-text alignment for this work that closely correlates with human perception.

