# OpenReview forum: "HarmonyLM: Advancing Unified Large-Scale Language Modeling for Sound and Music Generation"
_ICLR.cc/2024/Conference — ICLR 2024 Conference Withdrawn Submission_

### Official Review · Reviewer_PQPu · 2023-10-31

**Soundness:** 2 fair
**Presentation:** 2 fair
**Contribution:** 2 fair
**Rating:** 3
**Confidence:** 4

**Summary:**

This paper addresses the text-to-sound and text-to-music tasks using an LLM, an audio tokenizer and a vocoder.
The proposed method generates audio signal based on a small-scale LLM (flan-t5) that encodes a text, and SoundStream that encodes audio signals into audio tokens.
This paper appears well structured and easy to read. But I think important information is not adequatly provided to assess the quality of results.

**Strengths:**

Originality: Originality of problem definition would be limited because this paper addresses the existing tasks using existing benchmark datasets. But the experiments show that the perofmance is better than existing methods in many evaluation metrics. There would be some originality in some of technical details.

Quality: The proposed method excels in many evaluation metrics.

Clarity: The paper structure is clear and it would be easy to readers to find information they want.

Significance: The task of using pre-trained LLMs for different modalities is a major trend these days, and if the results are really convincing, it will attract many readers.

**Weaknesses:**

- While this is a trendy theme and there are many competitors, the originality of this method (that is, the difference from the existing methods), other than the evaluation metrics, has not been explicitly and sufficiently promoted.
- I am doubtful that the metrics are a legitimate measure of sound quality evaluation. Methods inspired by Frechet inception distance are often used, but in my experience they don't always match my actual feeling. The validity of KL is also not clear to me. Also, in any case, it would be nice if all the Tables have confidence intervals.
- The authods claim audio samples are available at github but I cannot access it.

**Questions:**

- I cannot access the github link and cannot assess the sound qualtiy.
- At least it would be good to have some spectrograms and some typical prompts (not templates.) More diagrams, equations, examples of prompts and spectrograms of the acoustic data would make this paper more convincing.
- As noted above, I am suspicious about the validity of the evaluation metrics. And I think confidence intervals (error bars) are also needed.
- I think the name "HamonyLM" is misleading. It sounds like that the method generates a chord sequence of music using an LLM.
- The figure shows that FLAN-T5 is frozen but the text says "During the training of language models" in 3.6.1. I do not see what does it mean.

---

### Official Review · Reviewer_Vp5Z · 2023-10-31

**Soundness:** 1 poor
**Presentation:** 2 fair
**Contribution:** 1 poor
**Rating:** 1
**Confidence:** 5

**Summary:**

The paper introduces HarmonyLM, a model both capable for text-to-sound and text-to-music generation tasks using discretized audio representations. It emphasizes model and data scalability and highlights achieving superior quality.

**Strengths:**

- **Paper Organization:** The authors have well-structured the paper into subsections, enhancing readability and enabling readers to easily locate to desired section.
- **Comparison with Previous Works:** The authors conducted a comparison with various previous methods on both text-to-sound and text-to-music tasks, providing a context for understanding the performance of proposed approach.

**Weaknesses:**

- **Lack of Novelty and Details**
    - The proposed method lacks novelty. It closely resembles the existing MusicGen [1] system with only minimal differences. The only difference from [1] is that they are using their own version of audio codec model and the training dataset.
    - The paper introduces their own version of audio codec model, but it lacks essential information for reproducibility, and there's no comparison with previous approaches like SoundStream or Encodec. Therefore, it's unclear if this module adds significant value to the proposed system, and confuses which module part actually contributes to the proposed system’s performance.
- **Evaluation Methodology**
    - Most critically, the paper's subjective evaluation methodology raises concerns. The MOS values for baseline systems are exactly the same as [2] and [1] for text-to-audio and text-to-music, respectively. This indicate the authors would have likely conducted the listening test just with their own samples; making the proposed approach’s result hard to justify.
    - Besides the inappropriate method of the subjective evaluation, it’s hard to know the contribution of this paper is whether on the training dataset or the system. There are open source models of the baseline systems that could be trained with the same training data, yet, the authors likely did not train those systems with their training dataset for a fair comparison.
- **Missing Information**
    - The paper references detailed information in the Appendix section, but the provided information are not sufficient for reproducing the proposed work..
    - The provided link for audio samples was not available at the time of review, limiting the subjective analysis and lacking the confidence for the reviewer.

**minor feedback**

- typo: @Contributions at the end of the Introduction: “We investigate the model and data scalability **~~if~~→of** large language models for both sound and music generation.”

[1] Copet, Jade, et al. "Simple and Controllable Music Generation." *Advances in Neural Information Processing Systems* (NeurIPS 2023).

[2] Huang, Jiawei, et al. "Make-An-Audio 2: Temporal-Enhanced Text-to-Audio Generation." *arXiv preprint arXiv:2305.18474* (2023).

**Questions:**

- Will the audio samples be publicly available, ensuring accessibility for future evaluations and comparisons?
- Will the authors provide implementation code for reproducibility?

---

### Official Review · Reviewer_GKXd · 2023-11-01

**Soundness:** 3 good
**Presentation:** 1 poor
**Contribution:** 1 poor
**Rating:** 1
**Confidence:** 4

**Summary:**

The authors describe HarmonyLM, a text-to-audio model that can generate both sound and music. The approach involves training an LLM on audio tokens extracted from large (audio, text) datasets using SoundStream. The authors compare their approach to strong baselines on a variety of text-to-audio benchmarks for sound and music.

**Strengths:**

The primary strength of this paper is the apparently strong quantitative and qualitative performance. The proposed method outperforms strong baselines on most evaluation metrics across both sound and music generation tasks.

**Weaknesses:**

Unfortunately, this paper has numerous issues preventing it from being of publishable quality. Most prominently, this paper is **poorly-written, lacking clarity and omitting important details**, and is **devoid of any salient research contribution**.

**Poorly-written**. This paper has many issues with presentational clarity and often omits key details of the proposed method. The clarity issues are apparent even at the very outset when describing the motivation and the contributions. This work is not the first to propose a unified method for generating both sound and music. In fact, one of their baselines, AudioLDM, is already capable of both tasks using a unified method. Additionally, the purported advantages of the HarmonyLM approach: “model scalability” and “data scalability”, are already features of basically every recent development in the field of audio generation. Moreover, the paper omits numerous crucial details. For example, the “unified prompt template” section appears to be the most salient departure from standard practice, but it is basically incomprehensible (I read it multiple times and still feel no closer to understanding how the approach works). Beyond the lack of high-level clarity and the omission of details, there are countless typos throughout (too many to list) and the paper is littered with handwavey unsubstantiated explanations (e.g., “extensive pre-training enables the models to learn new tasks effectively by leveraging in-context information and mimicking gradient descent through attention weights”).

**Lack of research contribution**. This paper essentially offers no substantive contribution to research. The methodology is more or less standard practice - HarmonyLM maps text tokens into audio tokens which are then converted into listenable audio, the same basic setup as much of the recent work in this area. The aspects of the methodology which are non-standard (e.g., vocoding tokens or the “unified prompt template”) are not described in sufficient detail to allow for reproduction. The paper boasts extremely impressive quantitative metrics, but the authors don’t discuss releasing the model, nor do they even provide any sound examples to listen to. How can anyone in the research community build on this work or even trust the results? Perhaps the list of datasets used to train this model would be useful for reproducing the results, but the identification of a set of text-audio datasets hardly constitutes a research contribution.

**Questions:**

- Can the authors provide sound examples for their method?
- Will the authors release the trained model?
- Can the authors elaborate on how the unified prompt template approach works and commit to refining the quality of the associated text?